# Prevalence and factors associated with multimorbidity among primary care patients with decreased renal function

Jennifer A. Hirst[1,2]*, José M. Ordóñez Mena[1,2], Chris A. O'Callaghan[3], Emma Ogburn[1], Clare J. Taylor[1,2], Yaling Yang[1,2], F. D. Richard Hobbs[1,2]

**1** Nuffield Department of Primary Care Health Science, University of Oxford, Radcliffe Observatory Quarter, Oxford, United Kingdom, **2** National Institute for Health Research (NIHR), Oxford Biomedical Research Centre, Oxford University Hospitals NHS Foundation Trust, Oxford, United Kingdom, **3** Nuffield Dept Medicine, University of Oxford, Oxford, United Kingdom

* jennifer.hirst@phc.ox.ac.uk

**Data Availability Statement:** Data cannot be shared publicly because of sensitive clinical data on potentially-identifiable human subjects. Data are

## Abstract

### Objectives

To establish the prevalence of multimorbidity in people with chronic kidney disease (CKD) stages 1–5 and transiently impaired renal function and identify factors associated with multimorbidity.

### Design and setting

Prospective cohort study in UK primary care.

### Participants

861 participants aged 60 and older with decreased renal function of whom, 584 (65.8%) had CKD and 277 (32.2%) did not have CKD.

### Interventions

Participants underwent medical history and clinical assessment, and blood and urine sampling.

### Primary and secondary outcome measures

Multimorbidity was defined as presence of $\geq$2 chronic conditions including CKD. Prevalence of each condition, co-existing conditions and multimorbidity were described and logistic regression was used to identify predictors of multimorbidity.

### Results

The mean ($\pm$SD) age of participants was 74$\pm$7 years, 54% were women and 98% were white. After CKD, the next most prevalent condition was hypertension (n = 511, 59.3%), followed by obesity (n = 265, 30.8%) ischemic heart disease (n = 145, 16.8%) and diabetes (n = 133, 15.4%). Having two co-existing conditions was most common (27%), the most

available from the University of Oxford (contact via email to: information.guardian@phc.ox.ac.uk) for researchers who meet the criteria for access to confidential data.

**Funding:** This work was supported in the form of funding by the National Institute for Health Research (NIHR) Oxford Biomedical Research Centre (BRC) awarded to JH and JOM, NIHR Community Healthcare MedTECH and In vitro Diagnostic Cooperative (MIC) awarded to DSL, and an NIHR Academic Clinical Lectureship awarded to CT. This work was also supported partially by the NIHR School for Primary Care Research (SPCR), the NIHR Collaboration for Leadership in Applied Research in Health and Care (CLARHC) Oxford, and the NIHR Oxford Biomedical Research Centre (OUHT BRC) for FDRH. This work is also partially supported by the NIHR Collaboration for Leadership in Applied Research in Health and Care (CLARHC) Oxford for RJM.

**Competing interests:** The authors have declared that no competing interests exist.

common combination of which was hypertension and obesity (29%). One or three conditions was the next most prevalent combination (20% and 21% respectively). The prevalence of multimorbidity was 73.9% (95%CI 70.9–76.8) in all participants and 86.6% (95%CI 83.9–89.3) in those with any-stage CKD. Logistic regression found a significant association between increasing age (OR 1.07, 95%CI 1.04–0.10), increasing BMI (OR 1.15, 95%CI 1.10–1.20) and decreasing eGFR (OR 0.99, 95%CI 0.98–1.00) with multimorbidity.

## Conclusions

This analysis is the first to provide an accurate estimate of the prevalence of multimorbidity in a screened older primary care population living with or at risk of CKD across all stages. Hypertension and obesity were the most common combination of conditions other than CKD that people were living with, suggesting that there may be multiple reasons for closely monitoring health status in individuals with CKD.

## Background

Chronic kidney disease (CKD) is a highly prevalent condition [1], affecting around 12% of the population globally. It is defined as decreased kidney function, measured as an estimated glomerular filtration rate (eGFR) of less than 60 ml/min/1·73m$^2$, or markers of kidney damage, present for at least 3 months duration [2,3]. As people age, there is a decline in eGFR [4–7], and the risk of poor health outcomes increases, including end stage renal disease, cardiovascular events and mortality [8–11]. Older people with CKD may also be living with other comorbidities as the prevalence of many long-term conditions increases with age [12]. It has been reported that CKD is rarely present without co-occurring diseases [13] and the prevalence of a different comorbidities in people with CKD have been described [14,15]. A study in UK primary care reported that 40% of people with stage 3 CKD are living with more than two chronic conditions and increased comorbidity was associated with reduced survival [16].

Healthcare delivery is generally centred on the treatment of single diseases, and multimorbidity, defined as having two or more comorbidities [12], is a huge challenge in the context of an aging population and optimal clinical management [17]. Over 50% of older people in the general population are living with more than one chronic condition [12] and, as populations age, the number of people with multimorbidity is predicted to double [18]. Multimorbidity increases the likelihood of hospital admission and length of stay, increases healthcare costs, reduces quality of life, increases adverse reaction due to polypharmacy [14,19,20] and increases mortality. Chronic disease management must therefore consider disease burden and how an individuals' various health problems may interact [21]. Multimorbidity in CKD carries a greater risk of dialysis [22], increased mortality [23] and incurs higher healthcare costs [24,25] compared to those with no comorbidities. It is therefore important to understand patterns between conditions in populations with multimorbidity and the clustering of diseases so that interventions can be targeted to treat people at highest risk.

The Oxford Renal Longitudinal Cohort Study (OxREN) is a prospective observational longitudinal cohort study in a primary care population aged 60 years and older. The study was established in 2013 in the central region of the southern United Kingdom. The objectives include defining the prevalence of selected risk factors for CKD and establishing the distribution of estimated kidney function and its trajectory over time in patients detected by a targeted

screening programme [26,27]. Understanding the range of comorbidities and the most common combinations of comorbidities will help the on-going clinical management of this population and establish whether people with CKD have higher rates of multimorbidity compared with the general population. Specifically, given the scarcity of data in the literature, we describe the prevalence of multimorbidity across the OxREN cohort and use data collected at the participants' baseline assessments to identify factors associated with multimorbidity.

## Methods

### Ethical approval and registration

The study protocol was approved by South Central Oxfordshire Research Ethics Committee B Reference 13/SC/0020 and registered with the UK Clinical Research Network (Registration number 36916).

### Study population and laboratory methods

OxREN recruited participants aged 60 years or above in primary care general practices in the Thames Valley region of the United Kingdom between 2013 and 2017. Interested and potentially eligible participants attended a primary visit where the study was explained and written informed consent was obtained. All OxREN participants who had a baseline assessment (n = 861) were included in the analysis [27]. Non-fasting blood and urine samples collected at the baseline study visit were analysed within 24 hours across two laboratories using identical albumin-creatinine ratio (ACR) and isotope dilution mass spectrometry (IDMS)-traceable enzymatic creatinine assays, and reported using the Modification of Diet in Renal Disease (MDRD) eGFR. Both samples for each patient were processed in the same laboratory. Based on laboratory results, 584 (65.8%) had a CKD diagnosis confirmed by two positive tests a minimum of 90 days apart [2]. The remaining 277 (32.2%) participants did not have a CKD diagnosis, but had one test with either a decreased eGFR ($<$60 ml/min/1.73m$^2$) or a raised urinary ACR ($>$3 mg/mmol), suggesting transiently impaired renal function, or may potentially arise from variability in analytical methods in those near diagnostic thresholds. More details of the population included in this analysis are given in our paper summarising the characteristics from the baseline study visit [28].

### Multimorbidity

Multimorbidity was defined as presence of $\geq$2 comorbidities at the baseline assessment as recommended by the Academy of Medical Sciences [29]. The comorbidities included were self-reported by the participant as part of the study, and confirmed from medical notes by a research nurse of 15 clinical conditions, which have been previously included in one or more measures of multimorbidity [12,30,31]. These were CKD, hypertension, diabetes, ischaemic heart disease, heart failure, atrial fibrillation, cerebrovascular disease, stroke, peripheral vascular disease, thyroid disease, anaemia, osteopenia, osteoporosis, obesity (BMI $\geq$ 30), and transient ischemic attack (TIA). These comorbidities were chosen for pragmatic reasons as they could be established using laboratory results or clinical records and were recorded during the baseline assessment of the OxRen study. In the primary analysis, CKD was determined by calculating eGFR using the MDRD equation [32]. However, the newer Chronic Kidney Disease Epidemiology Collaboration (CKD-EPI) equation was also used in a secondary analysis to calculate eGFR [33].

### Factors potentially associated with multimorbidity

Demographic, anthropometric, clinical and laboratory data collected at the baseline assessment were tested as potential predictor variables for multimorbidity. These predictors were

collected using self-administered questionnaires or via measurements taken by a practice or research nurse using standard procedures. The following predictor variables were assessed: increasing age (years), sex (male vs female), ethnicity (non-white vs white), increasing BMI (kg/m$^2$), increasing systolic and diastolic blood pressure (mm Hg), smoking status (former and current vs never), increasing alcohol intake (g/day), highest level of education (school vs higher), increasing eGFR (mg/min/1.73m$^2$) (estimated from serum creatinine using the Modification of Diet in Renal Disease (MDRD) equation [32]), increasing waist-to-hip ratio or having had a previous urinary tract infection. A sensitivity analysis was carried out to include stage of CKD, waist circumference and hip circumference (replacing waist-to-hip ratio). As an exploratory analysis we also explored whether cognitive function was associated with multimorbidity in 537 participants where this had been recorded.

## Statistical methods

The prevalence of each chronic condition was tabulated and presented in bar charts. These included prevalence of individual conditions, numbers living with a single condition or multiple conditions, and numbers with the most common comorbidities, for the full cohort and stratified by a diagnosis of CKD. Numbers of comorbid conditions were reported and tabulated by sex, whether participants had CKD and whether CKD was an existing diagnosis or identified from screening. Summary statistics are presented as mean and standard deviation and proportions and 95% confidence intervals (95%CI). Results were tabulated to show the prevalence of certain comorbidities across the full cohort, participants with CKD only, and those with stage 3 CKD to allow comparison with other primary care cohorts. Chi-squared tests were used to test differences in proportions. The R statistical software package "UpSetR" was used to identify all combinations of co-morbidities and rank them by frequency [34].

Results were presented as absolute prevalence and age-sex standardised prevalence using data from the Office for National Statistics [35]. Age and sex standardisation was carried out using direct standardisation methods [36]. Briefly, prevalence of multimorbidity in the OxREN cohort was calculated separately for males and females for each 5-year age group between 60–90 years and 90+ years. These were standardised to age-sex distribution of the English population as reported by the Office of National Statistics 2019 [35].

Univariable and multivariable logistic regression was used to estimate unadjusted and adjusted odds ratios (OR) and 95% confidence intervals (CI). Interactions were not considered. Normality of predictors were checked by visually inspecting histograms One person with missing BMI and blood pressure was excluded from the analysis. Stata version 16.0SE (StataCorp, Tx) and R (version 3.6.1) were used for statistical analyses.

## Results

The study population of 861 participants had a mean age of 74±7 years, 54% were women and they were predominantly white (98%). Mean baseline eGFR was 58±18 ml/min/1.73m$^2$ and 259 participants had a kidney function test result in the normal range, 34 had stage 1, 122 had stage 2, 379 had stage 3 and 67 had stage 4 CKD. The prevalence of KDIGO-criteria CKD was 65.8% (n = 584).

### Full cohort

After CKD, the most prevalent condition was hypertension, in 512 participants (59.5%), followed by obesity in 264 (30.7%), ischaemic heart disease in 146 (17.6%), diabetes in 133 (15.5%), atrial fibrillation in 110 (12.8%) and anaemia in 90 (10.5%). The absolute frequencies of all comorbidities and their prevalence and 95% CIs are presented in Table 1, both overall, and stratified by whether participants had CKD.

**Table 1. Comorbidities in the full dataset (n = 861).**

| Condition | Full-cohort (n = 861) | | CKD only (n = 584) | | No CKD (n = 277) | |
|---|---|---|---|---|---|---|
| | N | % (95% CI) | N | % (95% CI) | N | % (95% CI) |
| CKD | 584 | 67.8 (64.7 to 70.9) | 584 | 100 | 0 | 0 |
| Hypertension | 512 | 59.5 (56.2 to 62.7) | 374 | 64.0 (60.1 to 67.9) | 138 | 49.8 (43.9 to 55.7) |
| Obesity | 264 | 30.7 (27.5 to 33.7) | 174 | 29.8 (26.1 to 33.5) | 90 | 32.5 (27.0 to 38.0) |
| Ischaemic heart disease | 146 | 17.0 (14.5 to 19.5) | 113 | 19.4 (16.1 to 22.6) | 33 | 11.9 (8.1 to 15.7) |
| Diabetes | 133 | 15.5 (13.0 to 17.9) | 109 | 18.7 (15.5 to 21.8) | 24 | 8.7 (5.3 to 12.0) |
| Atrial fibrillation | 110 | 12.8 (10.5 to 15.0) | 84 | 14.4 (11.5 to 17.2) | 26 | 9.4 (6.0 to 12.8) |
| Thyroid disease | 106 | 12.3 (1.1 to 14.5) | 67 | 11.5 (8.9 to 14.1) | 39 | 14.1 (10.0 to 18.2) |
| Anaemia | 90 | 10.5 (8.4 to 12.5) | 71 | 12.2 (9.5 to 14.8) | 19 | 6.9 (3.9 to 9.8) |
| Cerebrovascular disease | 73 | 8.5 (6.6 to 10.3) | 59 | 10.1 (7.7 to 12.5) | 14 | 5.1 (2.5 to 7.6) |
| Osteoporosis | 64 | 7.4 (5.7 to 9.2) | 42 | 7.2 (5.1 to 9.3) | 22 | 7.9 (4.8 to 11.1) |
| Osteopenia | 45 | 5.2 (3.7 to 6.7) | 34 | 5.8 (3.9 to 7.7) | 11 | 4.0 (1.7 to 6.3) |
| TIA | 42 | 4.9 (3.4 to 6.3) | 33 | 5.7 (3.8 to 7.5) | 9 | 3.3 (1.2 to 5.3) |
| Heart failure | 39 | 4.5 (3.1 to 5.9) | 32 | 5.5 (3.6 to 7.3) | 7 | 2.5 (0.7 to 4.4) |
| Peripheral vascular disease | 29 | 3.4 (2.2 to 4.6) | 18 | 3.1 (1.7 to 4.5) | 11 | 4.0 (1.7 to 6.3) |
| Stroke | 17 | 2.0 (1.0 to 2.9) | 15 | 2.6 (1.3 to 3.9) | 2 | 0.7 (0 to 0.2) |

Numbers and percentages of participants living with multiple comorbidities are presented in Table 2, showing that 47.0% of people in the cohort were living with one to two conditions, which was similar for both men and women. Most people were living with two co-existing conditions (231, 26.8%), whereas 174 (20.2%) were living with one condition and 178 (20.7%) were living with three conditions. Only 51 people (5.9%) did not have any of the conditions.

The prevalence of multimorbidity (two or more coexisting conditions) in the full cohort was 73.9% (95%CI 70.9 to 76.8), rising slightly to 74.5% (95%CI 71.6 to 77.4) when standardised by age and sex to the general population (Table 3).

The most common combinations of comorbidities, other than CKD, in the full OxREN population are shown in Figs 1 and 2. Fig 1 shows that hypertension is the most common condition in people with up to three comorbidities, followed by obesity, diabetes and ischemic heart disease as the next most prevalent. For those with four or more comorbidities, numbers of people living with hypertension, diabetes, obesity and ischemic heart disease were similar.

## CKD versus non-CKD

Overall, 87% of people with CKD and 82% of people without CKD were living with at least one non-CKD condition. Those with CKD were most likely to have two or three co-existing

**Table 2. Prevalence of multimorbidity (including CKD) in the OxREN cohort stratified by sex and whether they had CKD.**

| Number of comorbidities | Full cohort (n = 861) | | Women (n = 468) | | Men (n = 393) | | CKD (n = 584) | | CKD (not including CKD co-morbidity) | No CKD (n = 277) | |
|---|---|---|---|---|---|---|---|---|---|---|---|
| | N | % (95%CI) | N | % (95%CI) | N | % (95%CI) | N | % (95%CI) | N, % (95%CI) | N | % (95%CI) |
| 0 | 51 | 6% (4–8) | 29 | 6% (4–8) | 22 | 6% (3–8) | 0 | - | 78, 13% (10–16) | 51 | 18% (14–23) |
| 1 | 174 | 20% (18–23) | 101 | 22% (18–25) | 73 | 19% (15–22) | 78 | 13% (11–16) | 154, 26% (22–30) | 96 | 35% (29–40) |
| 2 | 231 | 27% (24–30) | 127 | 27% (23–31) | 104 | 26% (22–31) | 154 | 26% (23–30) | 144, 25% (20–30) | 77 | 28% (23–33) |
| 3 | 178 | 21% (18–23) | 104 | 22% (18–26) | 74 | 19% (15–23) | 144 | 25% (21–28) | 100, 17% (13–21) | 34 | 12% (8–16) |
| 4 | 108 | 13% (10–15) | 49 | 10% (8–13) | 59 | 15% (11–19) | 100 | 17% (14–20) | 71, 12% (9–15) | 8 | 3% (1–5%) |
| 5 or more | 119 | 14% (12–16) | 58 | 12% (9–15) | 61 | 16% (12–19) | 108 | 18% (15–22) | 37, 6% (4–8) | 11 | 4% (2–6) |

**Table 3. Age- and sex-standardised prevalence of multimorbidity ($\geq$2 co-morbidities including CKD) in the OxREN cohort.**

| | Full cohort (n = 861) | | | CKD (n = 584) | | |
|---|---|---|---|---|---|---|
| | N | Prevalence (95% CI) | Age-sex standardised prevalence and 95% CI | N | Prevalence (95% CI) | Age-sex standardised prevalence and 95% CI |
| **Full cohort** | 636 | 73.9 (70.9 to 76.8) | 74.5 (71.6 to 77.4) | 506 | 86.6 (83.9 to 89.3) | 87.3 (84.6 to 90.0) |
| **Females** | 338 | 72.2 (68.2 to 76.3) | 74.3 (70.4 to 78.2) | 264 | 85.7 (81.8 to 89.6) | 86.2 (82.3 to 90.1) |
| **Males** | 298 | 75.8 (71.6 to 80.1) | 74.8 (70.5 to 79.1) | 242 | 87.7 (83.8 to 91.6) | 88.3 (84.6 to 92.0) |

conditions (26% and 25% respectively), meaning they were living with one or two conditions in addition to their CKD. Those without CKD were most likely to be living with only one condition (35%), followed by 28% with two co-existing conditions (Table 2).

More people with CKD were living with 3, 4 and 5 or more comorbidities compared with those without CKD. This remained true when CKD was not included as one of the conditions: 17% of people with CKD were living with 3 other conditions compared with 12% without CKD, but this difference was not significant (p = 0.067). A few individuals (0.4%) were living with eight comorbidities.

When results were stratified by whether participants had CKD when they entered the cohort, or whether they were identified to have CKD on screening, people with existing CKD had overall more comorbidities than those with newly diagnosed CKD (p = 0.034), and tended to be living with more co-existing comorbidities than those with newly-diagnosed CKD (S2 Table). The proportion of people with hypertension was greater in people with CKD (n = 374, 64%) compared to those without CKD (n = 138, 50%) in those with up to three comorbidities (in addition to CKD) (S3 Fig).

The prevalence of multimorbidity in those with CKD, was 86.6% (95%CI 83.9 to 89.3), which increased to 87.3% (95%CI 84.6 to 90.0) with age and sex adjustment when standardised

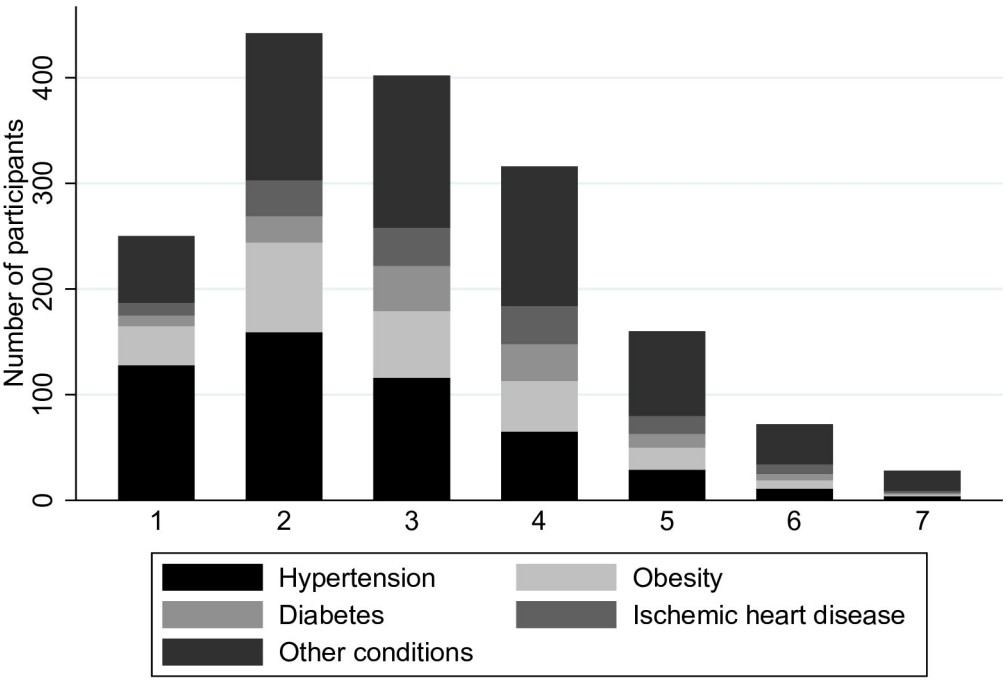

**Fig 1. Number of people with the four most common conditions stratified by number of comorbidities (not including CKD) in the full OxRen population.**

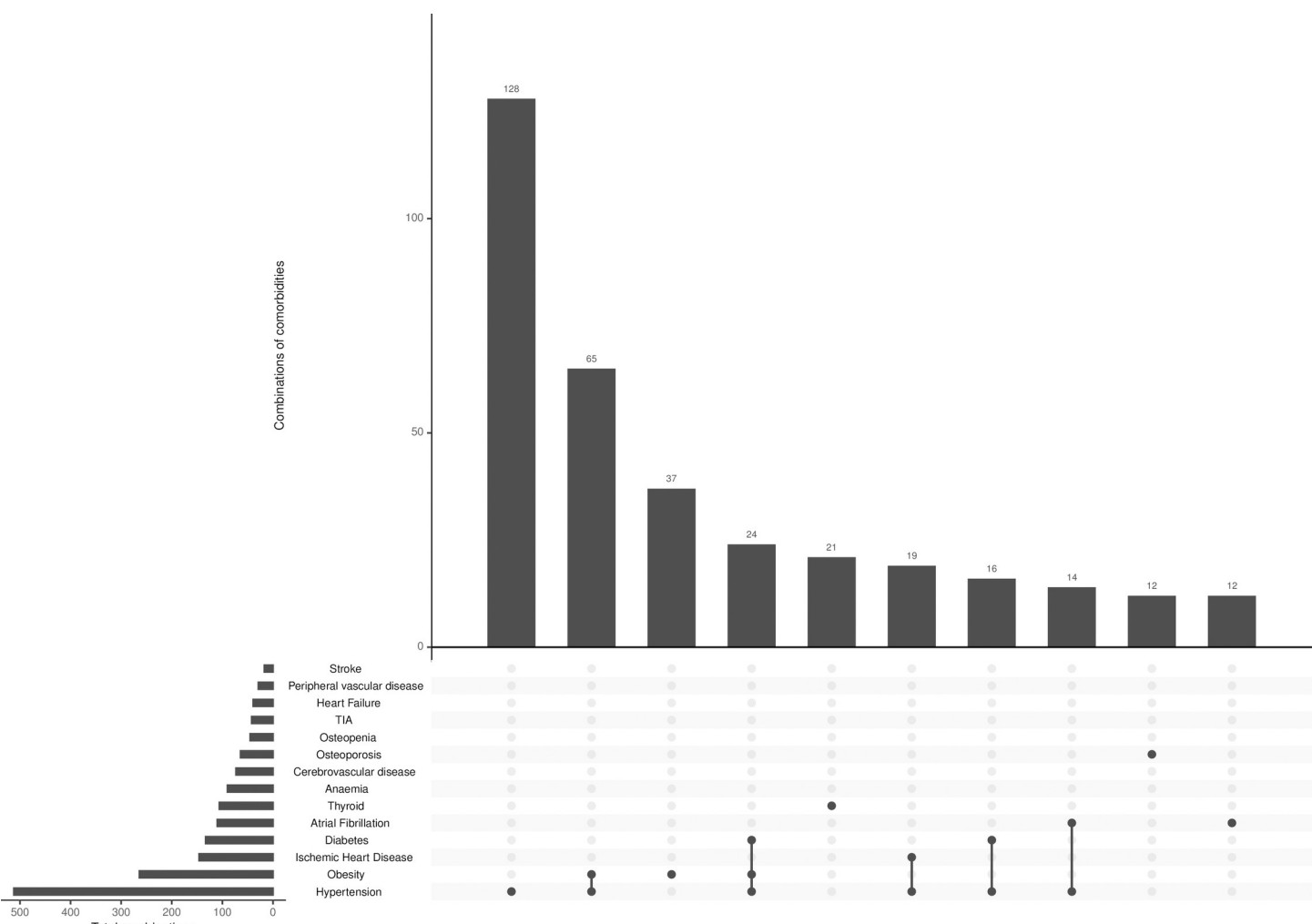

**Fig 2. Ten most common comorbidities and combinations (excluding CKD) in the full OxRen population.**

by age and sex to the general population (Table 3). S1 Fig shows absolute numbers of people living with one to eight comorbidities stratified by whether or not they have CKD.

Fig 2 shows the ten most prevalent comorbidities other than CKD, either present as a single condition or in combination with other conditions. Although over 500 people have hypertension, 126 people have hypertension alone (possibly in addition to CKD), 66 have a combination of obesity and hypertension, and 37 people have obesity alone. S1 Table shows prevalence of each individual comorbidity in the OxREN cohort overall, in those with CKD and those with stage 3 CKD to allow comparison with another study [16] which recruited participants with stage 3 CKD only. Prevalence of multimorbidity was slightly higher (p = 0.018) among those with existing CKD and a higher proportion of people were living with more comorbidities compared with newly diagnosed CKD (S2 Table).

## Differences by GFR-estimating equation and sex

When CKD-EPI was used to determine eGFR, there were slightly more people with two and three comorbidities compared with MDRD (S2 Fig).

**Table 4. Logistic regression showing predictors of multimorbidity ($\geq$2 co-morbidities including CKD).**

| Covariate | N | MM | Univariable analysis, OR and 95% CI | P-value | Multivariable analysis, OR and 95% CI | P-value |
|---|---|---|---|---|---|---|
| Age (per 1 year) | 861 | 635 | 1.048 (1.024 to 1.073) | <0.0001 | 1.068 (1.039 to 1.097) | <0.0001 |
| Women | 468 | 338 | 1.00 (reference) | | 1.00 (reference) | |
| Men | 393 | 298 | 1.206 (0.888 to 1.640) | 0.231 | 0.675 (0.422 to 1.079) | 0.100 |
| BMI (per 1 kg/m$^2$) | | | 1.142 (1.101 to 1.184) | <0.0001 | 1.148 (1.099 to 1.199) | <0.0001 |
| Smoking status | | | | | | |
| • Never smoker | 467 | 329 | 1.00 (reference) | | 1.00 (reference) | |
| • Former smoker | 356 | 278 | 1.352 (0.623 to 2.930) | 0.445 | 1.646 (0.706 to 3.839) | 0.249 |
| • Current smoker | 38 | 29 | 1.495 (1.086 to 2.059) | 0.014 | 1.331 (0.936 to 1.893) | 0.111 |
| Alcohol dose (per 1 g/day) | | | 1.003 (0.988 to 1.019) | 0.680 | 1.010 (0.991 to 1.029) | 0.307 |
| Secondary education | 657 | 494 | 1.00 (reference) | | 1.00 (reference) | |
| Higher education | 204 | 142 | 0.756 (0.534 to 1.069) | 0.114 | 1.044 (0.708 to 1.539) | 0.828 |
| Systolic blood pressure (per 1 mmHg) | | | 1.002 (0.994 to 1.010) | 0.591 | 0.997 (0.985 to 1.009) | 0.626 |
| Diastolic blood pressure (per 1 mmHg) | | | 0.996 (0.982 to 1.009) | 0.523 | 0.996 (0.977 to 1.015) | 0.672 |
| White ethnicity | 846 | 627 | 1.00 (reference) | | 1.00 (reference) | |
| Non-white ethnicity | 14 | 9 | 0.632 (0.209 to 1.905) | 0.415 | 0.897 (0.266 to 3.019) | 0.860 |
| Waist: hip ratio | | | 55.6 (10.0 to 308.3) | <0.0001 | 5.378 (0.402 to 72.021) | 0.204 |
| eGFR (per 1 mg/min/1.73m$^2$) | | | 0.979 (0.971 to 0.988) | <0.0001 | 0.988 (0.978 to 0.997) | 0.010 |
| Urinary tract infection (Previous diagnosis compared with never diagnosed) | | | 0.728 (0.536 to 0.989) | 0.042 | 0.754 (0.528 to 1.079) | 0.123 |

Prevalence of multimorbidity was slightly higher in men than in women after adjustment for age and sex but these differences were not significant (75.0% for men versus 73.9% for women in the full cohort, p = 0.231, and 88.4% for men versus 85.9% for women in those with CKD, p = 0.486).

## Predictors of multimorbidity

The results of the univariable and multivariable logistic regression analyses to identify predictors of multimorbidity are show in Table 4. In the univariable analysis, increasing age, BMI or waist-to-hip ratio, decreasing eGFR or being a current smoker versus having never smoked were associated with significantly higher odds of having multimorbidity, whereas having a previous urinary tract infection was associated with lower odds of having multimorbidity. In the multivariable analysis adjusting for all other predictors, only increasing age, increasing BMI and decreasing eGFR remained significant. Applying a significant level of 0.004 to correct for testing multiple variables, only age and BMI remained significant predictors of multimorbidity. The sensitivity analysis using different covariates showed similar results (S3 Table). The exploratory analysis did not find any association between cognitive function and multimorbidity (OR 4.988, 95%CI 0.926 to 1. 043).

## Discussion

### Summary

This analysis has established that 87% of people with CKD were living with at least one other comorbidity. Hypertension was the most common co-morbid condition in CKD with a prevalence of 64%. In comparison, 50% of people without CKD were living with hypertension and 82% were living with at least one long-term health condition. After CKD and hypertension, obesity, ischemic heart disease and diabetes were the next most common co-morbidities.

More people with CKD were living with three or more comorbidities (the most common combination being hypertension, obesity and diabetes) compared to those without CKD, which remained true even when CKD was not included as one of the conditions. The age-sex standardised prevalence of multimorbidity was 75% for the whole population and 87% in those with CKD. Some people (0.4%) were living with as many as eight coexisting comorbidities. In an analysis to determine predictors of multimorbidity, age, BMI and declining eGFR were the only factors significantly associated with multimorbidity after adjustment for other covariates.

## Strengths and limitations

This analysis is the first to provide an accurate estimate of the prevalence of multimorbidity in a screened older primary care population living with or at risk of CKD across all stages. It provides reliable results through prospectively collected data and medical records to collect anthropometric measurements and demographic characteristics and determine the presence and number of co-existing conditions. Furthermore, we have reported the prevalence of different comorbidities and their combinations to describe in depth the range of conditions people with chronic kidney disease are living with.

This work has some limitations. Our cohort was recruited in the generally affluent parts of the UK and may therefore not fully reflect the socioeconomic and ethnic diversity of the UK population [27,37]. Because this cohort may have a higher socioeconomic status than the general population, it is possible that we have underestimated the prevalence of some conditions or multimorbidity in general. This may therefore mean that these data provide conservative estimates compared with populations of a similar age in all parts of the UK. To be transparent about the generalisability, we have discussed the results in the context of other studies, both in the UK and internationally. We did not collect data on cancer as people with terminal illness were excluded from the study, but this has not limited our ability to make comparisons with other cohorts, which similarly excluded people with terminal illness [16].

## Comparison with existing literature

Previous studies in UK primary care have also reported hypertension to be the most common comorbidity in populations with CKD [16,38]. The Renal Risk in Derby study [16] found a prevalence of hypertension of 87.8% in people with stage 3 CKD, which is unsurprisingly higher than the 64% in the OxREN CKD population, as our screening identified earlier stages of CKD. They also observed a higher prevalence of anaemia (24% versus 12% in OxREN participants with CKD), which has been reported as a major complication of CKD and is associated with poor outcomes [39]. The prevalence of diabetes, thyroid disorder, ischemic heart disease, cerebrovascular disease, peripheral vascular disease and heart failure in the Renal Risk in Derby study was similar to that in our cohort. Mean age and ethnicity of populations were similar in both studies, but there may have been differences in socioeconomic status between the cohorts, which could have accounted for differences in prevalence in hypertension and anaemia, which remained even when our analysis was restricted to those with stage 3 CKD. Prevalence of multimorbidity in the OxREN population was slightly higher than the 67% reported in a general Scottish primary care population over 65 years [12]. Another UK-wide cohort reported a prevalence of hypertension of 42% in those with CKD stages 1–2, and 64% in those with CKD stages 3–5 [40]. They also found a prevalence of diabetes of 13% in those with CKD stages 1–2 and 19% in those with CKD stages 3–5. These were similar to our findings, but their analysis included anyone over 18 years of age, so was not restricted to an older population.

Internationally, primary care may differ from the UK, making it difficult to make true comparisons. Other cohorts have generally found higher prevalence of comorbidities in people with

CKD. The SCOPE study described patterns of multimorbidity in geriatric and nephrology outpatients aged over 75 years across several European countries [13]. They reported CKD and hypertension to be the most common combination of comorbidities with a prevalence of 54%, lower than the 64% in the OxREN cohort. They reported a similar prevalence of diabetes to OxREN, but a much higher prevalence or osteoporosis compared with our cohort, which may reflect the older age of participants. In an Australian cohort, the prevalence of hypertension was 91% and diabetes 44% [41], but recruitment took place in a renal clinic and CKD stages were predominantly 3b-4, suggesting more severe disease than our cohort. A randomised trial recruiting older people from the US and Australia found that 83% of those with CKD had hypertension and 15% had diabetes [42]. These results present a similar prevalence of diabetes and a higher prevalence of hypertension compared with our cohort, despite participants being a similar age.

Ten of the 15 comorbidities assessed in the OxREN cohort were concordant with CKD, meaning they have similar pathophysiologies or recommended treatments. Discordant conditions are unrelated and treatment for one condition may have a negative impact on kidney function [43]. Not all comorbidities have the same impact on clinical decision-making or patients' ability to self-manage chronic diseases, and it has been suggested that the presence of comorbidities with different pathologies are associated with increased risk for adverse health outcomes. This may be because healthcare is often not integrated, meaning that people living with multiple coexisting conditions may receive medications for one condition, which are contra-indicated for one of their other conditions [43]. The majority of the comorbidities in our cohort are associated with poor health outcomes.

Increasing age, being a smoker, higher BMI and lower eGFR have been reported to be significantly associated with multimorbidity in people with stage 3 CKD [16]. One study of 400 participants found a significant association between age and decreasing cognitive function with incident multimorbidity in a population aged over 75 years [44]. An exploratory analysis in the OxREN cohort did not find any association between cognitive function and multimorbidity at baseline. Socioeconomic status has also been reported to be associated with multimorbidity in the general population [45]. We did not examine this directly in our analysis, but we found no association between education status and multimorbidity.

## Clinical implications

The scale of comorbidities we have found in this analysis suggest that a patient-centred approach to treatment and care is required. This is in line with recommendations from the UK's National Institute for Heath and Care Excellence (NICE) [21] and Department of Health [46] in treating multimorbidity. The high proportion of people in our cohort confirmed to be living with hypertension provided a further imperative for regular blood pressure monitoring to guide treatment to optimal blood pressure targets with antihypertensives. Although guidelines recommend blood pressure treatment thresholds and medication regimens for people with CKD [2,3], they do not suggest a frequency for blood pressure monitoring. Because of the high prevalence of hypertension in those with CKD, clinicians should consider frequent blood pressure monitoring to reduce cardiovascular risk [47]. Our finding that hypertension and obesity were the most common combinations of conditions in addition to CKD that people were living with, suggests that there may be multiple reasons for closely monitoring health status in individuals with CKD.

## Supporting information

**S1 Fig. Bar chart showing numbers of OxREN participants with between 1–8 comorbidities, including CKD, stratified by whether or not participants has CKD (CKD established**

**using the CKD-EPI equation).**
(DOCX)

**S2 Fig. Bar chart showing numbers of OxREN participants with between 1–8 comorbidities including CKD, stratified by whether or not participants has CKD (CKD established using the CKD-EPI equation).**
(DOCX)

**S3 Fig. Comparison of combinations of comorbidities other than CKD in participants with CKD (CKD cohort, n = 584) and participants without CKD but have had a test suggesting that they have transiently impaired kidney function (non-CKD cohort, n = 277).**
(DOCX)

**S1 Table. Compare prevalence of comorbidities in OxRen, those with CKD in OxRen, those with stage 3 CKD in OxRen and RRID (Renal Risk in Derby) study(1), a cohort of 1741 participants with stage 3 CKD.**
(DOCX)

**S2 Table. Prevalence of multimorbidity in people with CKD in the OxREN cohort stratified by whether their CKD was existing or newly diagnosed.**
(DOCX)

**S3 Table. Logistic regression to identify predictors of multimorbidity, including stage of CKD and waist circumference and hip circumference instead or waist-to-hip ratio in the analysis.**
(DOCX)

## Acknowledgments

Thank you to the OxREN Steering Committee members: Professor Maarten Taal, Professor of Medicine, Royal Derby Hospital & University of Nottingham (chair), Dr Simon Fraser, GP, Dr Marion Judd (patient representative), Dr Elizabeth Holloway (patient representative). Staff in Oxford University's Primary Care Clinical Trials Unit were responsible for trial management (Dr Hannah Swayze and Ms Rebecca Lowe), database development, data entry and cleaning (Dr Sue Smith) and patient recruitment (Heather Rutter).

## Author Contributions

**Conceptualization:** Chris A. O'Callaghan, Clare J. Taylor, F. D. Richard Hobbs.

**Data curation:** Jennifer A. Hirst, Emma Ogburn.

**Formal analysis:** Jennifer A. Hirst, José M. Ordóñez Mena.

**Funding acquisition:** Chris A. O'Callaghan, Clare J. Taylor, F. D. Richard Hobbs.

**Investigation:** Jennifer A. Hirst, José M. Ordóñez Mena, Emma Ogburn.

**Methodology:** Jennifer A. Hirst, José M. Ordóñez Mena, Clare J. Taylor.

**Project administration:** Jennifer A. Hirst, Emma Ogburn, F. D. Richard Hobbs.

**Supervision:** Jennifer A. Hirst, Chris A. O'Callaghan, Clare J. Taylor, F. D. Richard Hobbs.

**Validation:** José M. Ordóñez Mena.

**Visualization:** Chris A. O'Callaghan, Clare J. Taylor, F. D. Richard Hobbs.

**Writing – original draft:** Jennifer A. Hirst.

**Writing – review & editing:** Jennifer A. Hirst, José M. Ordóñez Mena, Chris A. O'Callaghan, Emma Ogburn, Clare J. Taylor, Yaling Yang, F. D. Richard Hobbs.

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
