## [Decision Letter · Decision Letter 0]

19 Nov 2020

PONE-D-20-32945

Prevalence and factors associated with multimorbidity among primary care patients with decreased renal function.

PLOS ONE

Dear Dr. Hirst,

Thank you for submitting your manuscript to PLOS ONE. After careful consideration, we feel that it has merit but does not fully meet PLOS ONE’s publication criteria as it currently stands. Therefore, we invite you to submit a revised version of the manuscript that addresses the points raised during the review process.

**The manuscript focuses on an interesting topic. The study, however, presents some important shortcomings that preclude at this stage to draw meaningful conclusions, and should be addressed. To mention few of them, i) need to describe laboratory methodology, i.e. serum creatinine assay and its traceability to IDMS and authors’ methodology for detecting raised ACR of > 3 mg/mmol; ii) unclear the authors’ sampling methodology; iii) concern about the fact that transiently impaired kidney function was not defined nor they have indicated baseline measures or the magnitude of the transient impairment; iv) need to present results with subheadings for better reading; v) need to compare authors’ data locally with national datasets as well as international registry reports like Australian, Japanese, Germany, or European CKD data, and discuss the implications in detail.**

We look forward to receiving your revised manuscript.

Kind regards,

Giuseppe Remuzzi

Academic Editor

PLOS ONE

Journal Requirements:

2. Please provide additional details regarding participant consent. In the ethics statement in the Methods and online submission information, please ensure that you have specified (i) whether consent was informed and (ii) what type you obtained (for instance, written or verbal, and if verbal, how it was documented and witnessed). If your study included minors, state whether you obtained consent from parents or guardians. If the need for consent was waived by the ethics committee, please include this information.

Reviewers' comments:

Reviewer's Responses to Questions

**Comments to the Author**

1. Is the manuscript technically sound, and do the data support the conclusions?

Reviewer #1: Yes

Reviewer #2: Partly

2. Has the statistical analysis been performed appropriately and rigorously? 

Reviewer #1: Yes

Reviewer #2: Yes

3. Have the authors made all data underlying the findings in their manuscript fully available?

Reviewer #1: Yes

Reviewer #2: No

4. Is the manuscript presented in an intelligible fashion and written in standard English?

Reviewer #1: Yes

Reviewer #2: Yes

5. Review Comments to the Author

Reviewer #1: I congratulate the authors for such a nice study. It's important to delineate the burden of CKD and multi-morbidity at primary care level.

I have few suggestions

1. The presentation of results could be sub-headings for better reading. For example if the results are presented as Overall cohort, CKD, Non CKD and Male and female. I found that there was lot of information, especially % all over the place and I moved backwards and forwards to read.

2. This data from primary care is very important and actually define a more milder cohort of CKD. If you take an average 74 year old white (98%) UK resident with hypertension, obesity and CKD as the top co-morbidities with rest of the conditions at much lower percentages including diabetes, cardiac and other vascular conditions, as seen in your data, this group represents a milder form of disease compared to any group of CKD patients seen in speciality renal practices. For example in out cohort in Australia ( 90% Caucasians) seen in specialist clinics, Obesity is >50%, Diabetes is 44%, hypertension 91%, coronary artery disease 25% and other vascular complications in the range of 10%. (https://doi.org/10.1111/nep.13567). Similar is the case in other CKD registry reports. The implications of these comparisons with other international cohorts are huge. Either you evaluated a milder form of disease or your primary care programs are working very well so much so the CKD patients are managed very well at primary care level and mulitmorbidity profile is much contained and better. I suggest that you compare your data locally with national datasets as well as international registry reports like Australian, Japanese, German or European CKD data. I think your data send a much powerful message regarding the success of primary care in your country.

Reviewer #2: Thank you for the opportunity to review this paper.

There are methodological issues which I believe should be addressed.

One such issue you describe in your limitations (your population is not representative of the UK population and findings cannot therefore be extrapolated to the general population).

You must describe laboratory methodology ie serum creatinine assay and its traceability to IDMS and your methodology for detecting a raised ACR of > 3mg/mmol.

You must also describe your sampling methodology - when were blood samples taken for serum creatinine analysis, were they fasted or non-fasted (relationship to a protein containing meal); when were urine samples for ACR testing taken; were samples stored before analysis.

You have not defined transiently impaired kidney function nor have you indicated baseline measures or the magnitude of the transient impairment.

Although correct to say that any test has both physiological and analytical variability without detailing the conditions under which the test sample is taken, the assay used or the magnitude of the abnormal result it is impossible to draw meaningful conclusions.

6. PLOS authors have the option to publish the peer review history of their article (what does this mean?). If published, this will include your full peer review and any attached files.

Reviewer #1: **Yes: **Sree K Venuthurupalli

Reviewer #2: **Yes: **P E Stevens

---

## [Author Response · Author response to Decision Letter 0]

10 Dec 2020

Reviewer #1: 

I congratulate the authors for such a nice study. It's important to delineate the burden of CKD and multi-morbidity at primary care level.

I have few suggestions 

Thank you for your comments

1. The presentation of results could be sub-headings for better reading. For example if the results are presented as Overall cohort, CKD, Non CKD and Male and female. I found that there was lot of information, especially % all over the place and I moved backwards and forwards to read. We have restructured the results to findings in the full cohort from the CKD and non-CKD populations.

We have also included another subheading: “Differences by GFR-estimating equation and sex” to make comparisons between the MDRD and CKD-EPI equations and males and females.

2. This data from primary care is very important and actually define a more milder cohort of CKD. If you take an average 74 year old white (98%) UK resident with hypertension, obesity and CKD as the top co-morbidities with rest of the conditions at much lower percentages including diabetes, cardiac and other vascular conditions, as seen in your data, this group represents a milder form of disease compared to any group of CKD patients seen in speciality renal practices. For example in out cohort in Australia ( 90% Caucasians) seen in specialist clinics, Obesity is >50%, Diabetes is 44%, hypertension 91%, coronary artery disease 25% and other vascular complications in the range of 10%. (https://doi.org/10.1111/nep.13567). Similar is the case in other CKD registry reports. The implications of these comparisons with other international cohorts are huge. Either you evaluated a milder form of disease or your primary care programs are working very well so much so the CKD patients are managed very well at primary care level and mulitmorbidity profile is much contained and better. I suggest that you compare your data locally with national datasets as well as international registry reports like Australian, Japanese, German or European CKD data. I think your data send a much powerful message regarding the success of primary care in your country. Thank you for your comments. 

We have changed the discussion to include a comparisons with literature, first in the UK followed by international settings.

This section in the Discussion has now been substantially revised.

Reviewer #2: 

Thank you for the opportunity to review this paper.

There are methodological issues which I believe should be addressed.. 

One such issue you describe in your limitations (your population is not representative of the UK population and findings cannot therefore be extrapolated to the general population).

 The reviewer has pointed out a concern about the generalisability. We have compared our results with those from other studies and suggested reasons for differences in prevalence of multimorbidity and individual comorbidities. We have included a sentence to explain that we have compared with other studies: “To be transparent about the generalisability, we have discussed the results in the context of other studies, both in the UK and internationally.”

You must describe laboratory methodology ie serum creatinine assay and its traceability to IDMS and your methodology for detecting a raised ACR of > 3mg/mmol.

 We have clarified these points in the text: “Non-fasting blood and urine samples collected at the baseline study visit were analysed within 24 hours across two laboratories using identical albumin-creatinine ratio (ACR) and isotope dilution mass spectrometry (IDMS)-traceable enzymatic creatinine assays, and reported using the Modification of Diet in Renal Disease (MDRD) eGFR. Both samples for each patient were processed in the same laboratory. The remaining 277 (32.2%) participants did not have a CKD diagnosis, but had one test with either a decreased eGFR (<60 ml/min/1.73m2) or a raised non-fasting urinary ACR (>3 mg/mmol), suggesting transiently impaired renal function, or may potentially arise from variability in analytical methods in those near diagnostic thresholds.”

You must also describe your sampling methodology - when were blood samples taken for serum creatinine analysis, were they fasted or non-fasted (relationship to a protein containing meal); when were urine samples for ACR testing taken; were samples stored before analysis.

 Please see above

You have not defined transiently impaired kidney function nor have you indicated baseline measures or the magnitude of the transient impairment.

 The participants we classified as having transiently impaired renal function had a single eGFR or ACR test over the diagnostic threshold, but did not meet the KDIGO criteria for sustained impairment. We have included these participants in our cohort as we believe that they may be more likely to progress to CKD than the general population. We have removed most references to transiently impaired renal function in the text and instead refer to this population as non-CKD. 

Details of this population, as well as other characteristics of the participants are given in a paper published earlier this year. Rather than repeat these, we have referred the reader to the paper: “More details of the population included in this analysis are given in our paper summarising the characteristics from the baseline study visit.(28)” 

Although correct to say that any test has both physiological and analytical variability without detailing the conditions under which the test sample is taken, the assay used or the magnitude of the abnormal result it is impossible to draw meaningful conclusions 

We hope the more detailed methods given above have served to clarify this.

---

## [Decision Letter · Decision Letter 1]

23 Dec 2020

Prevalence and factors associated with multimorbidity among primary care patients with decreased renal function.

PONE-D-20-32945R1

Dear Dr. Hirst,

We’re pleased to inform you that your manuscript has been judged scientifically suitable for publication and will be formally accepted for publication once it meets all outstanding technical requirements.

**The revised manuscript is definitely improved. The authors have properly addressed all the points and comments raised by the reviewers. **

Kind regards,

Giuseppe Remuzzi

Academic Editor

PLOS ONE

Additional Editor Comments (optional):

Reviewers' comments:

Reviewer's Responses to Questions

**Comments to the Author**

1. If the authors have adequately addressed your comments raised in a previous round of review and you feel that this manuscript is now acceptable for publication, you may indicate that here to bypass the “Comments to the Author” section, enter your conflict of interest statement in the “Confidential to Editor” section, and submit your "Accept" recommendation.

Reviewer #1: All comments have been addressed

Reviewer #2: All comments have been addressed

2. Is the manuscript technically sound, and do the data support the conclusions?

Reviewer #1: Yes

Reviewer #2: (No Response)

3. Has the statistical analysis been performed appropriately and rigorously? 

Reviewer #1: (No Response)

Reviewer #2: (No Response)

4. Have the authors made all data underlying the findings in their manuscript fully available?

Reviewer #1: Yes

Reviewer #2: (No Response)

5. Is the manuscript presented in an intelligible fashion and written in standard English?

Reviewer #1: Yes

Reviewer #2: (No Response)

6. Review Comments to the Author

Reviewer #1: Thank you very much for addressing my feedback and comments. The current manuscript is definitely an improved version. I have no further comments

Reviewer #2: (No Response)

7. PLOS authors have the option to publish the peer review history of their article (what does this mean?). If published, this will include your full peer review and any attached files.

Reviewer #1: **Yes: **Sree Krishna Venuthurupalli

Reviewer #2: No

---

## [Editor Report · Acceptance letter]

6 Jan 2021

PONE-D-20-32945R1 

Prevalence and factors associated with multimorbidity among primary care patients with decreased renal function. 

Dear Dr. Hirst:

I'm pleased to inform you that your manuscript has been deemed suitable for publication in PLOS ONE. Congratulations! Your manuscript is now with our production department. 

Kind regards, 

on behalf of

Prof. Giuseppe Remuzzi 

Academic Editor

PLOS ONE